# Study on Curing Deformation of Composite Thin Shells Prepared by M-CRTM with Adjustable Injection Gap

**DOI:** 10.3390/polym14245564

**Published:** 2022-12-19

**Authors:** Ce Zhang, Ying Sun, Jing Xu, Xiaoping Shi, Guoli Zhang

**Affiliations:** 1School of Textile Science and Engineering, Tiangong University, Tianjin 300387, China; 2Ministry of Education Key Laboratory of Advanced Textile Composite Materials, Institute of Composite Materials, Tiangong University, Tianjin 300387, China; 3AVIC Aerospace Life-Support Industries, Ltd., Xiangyang 441003, China

**Keywords:** composites, thin shell, curing deformation, RTM, M-CRTM, porosity

## Abstract

A composite thin shell with a high fiber volume fraction prepared by resin transfer molding (RTM) may have void defects, which create deformations in the final curing and lead to the final product being unable to meet the actual assembly requirements. Taking a helmet shell as an example, a multi-directional compression RTM (M-CRTM) method with an adjustable injection gap is proposed according to the shape of the thin shell. This method can increase the injection gap to reduce the fiber volume fraction during the injection process, making it easier for the resin to penetrate the reinforcement and for air bubbles to exit the mold. X-ray CT detection shows that the porosity of the helmet shell prepared by the newly developed technology is 36.6% lower than that of the RTM-molded sample. The void’s distribution is more uniform, and its size is decreased, as is the number of voids, especially large voids. The results show that the maximum curing deformation of the M-CRTM-molded helmet shell is reduced by 13.7% compared to the RTM molded sample. This paper then further studies the deformation types of the shell and analyzes the causes of such results, which plays an important role in promoting the application of composite thin shells.

## 1. Introduction

Carbon-fiber-reinforced polymer (CFRP) has an excellent specific strength and specific modulus, and the composite thin shell is applied in the aerospace field for applications such as solid rocket motors and pilot helmets. These fields require a high precision and a small dimensional tolerance for composite parts due to the limited assembly tolerance. If curing deformations occur after shell demolding, this may exceed the assembly tolerance and affect the assembly of the composite shell. The forced assembly causes internal stress in the parts, resulting in the scrapping of parts and a huge amount of manpower and material resources being wasted. Therefore, in the manufacturing process, it is necessary to control the molding process to reduce curing deformations in the composite shell.

In the curing process of thermosetting composites, the curing process parameters will affect the residual stress [1,2] and the microstructure [3], which in turn influence curing deformations. The main factors that affect curing deformation for composites include the resin’s chemical shrinkage coefficient, the layup angles, the stacking sequences, tool–part interaction, etc. [4]. In order to reduce curing deformations, many methods have been developed, such as symmetrical layup [5] or similar isotropic stack sequences [6], improving the fiber volume fraction [7], the nano-modification of resin [8], tool–part contact optimization [9], applying prestress on a layer or part of a layer [10], etc. In addition to the above factors, the technology used in the preparation of composites has an effect on the curing deformation. The manufacturing process directly determines the forming quality of the parts. For example, the existence of void defects will lead to increased stress concentration [11] and in turn affect curing deformation. On the other hand, voids are one of the most common manufacturing defects in composites. For high-quality and high-performance aerospace products, small differences in voids are also very important. It is well known that porosity has an adverse effect on the strength of composite laminates, especially in terms of matrix-dominated mechanical properties such as compressive strength [12] and flexural strength [13]. For example, Zhang [14] prepared three samples with different voids (0.33~1.50%) using different autoclave pressures during one curing cycle. The experimental results showed that the compressive strength, bending strength, and ILSS decreased with the increase in porosity. Liu [15] used different curing pressures to induce the preparation of composite laminates with voids ranging from 0 to 3.5%. The results showed that both the tensile modulus and the flexural modulus decreased with the increase in porosity. Thanh [16] studied the free vibration and buckling problems of annular plates and conical and cylindrical shells with porosity-free top surfaces and porosity-rich bottom surfaces. The author found that the impact of porosity leads to a decrease in structural stiffness, so the frequencies and buckling load decrease with the increase in the porosity coefficient. Subsequently, the effects of porosity distributions on the linear and nonlinear bending responses of porous material nanoplates graded using the sigmoid function were investigated. The results showed that symmetric and asymmetric porosity have significant effects on a material’s bending properties [17]. Through DIC system measurements and finite element simulation, Wang [18] found that the presence of void defects affects the strain field and produces a stress-concentration phenomenon. Therefore, it is important to obtain the real geometry of the void defects present to predict the strength and analyze the security of a structure. In addition, voids occurring inside of parts affect the curing shrinkage of the resin in the corresponding area [19] and thus affect the curing deformation of the composite. Mesogitis [11] believed that voids would affect the distribution of the fiber volume fraction, further affecting the thermal, mechanical, and thermomechanical properties of components and introducing variability into the curing of composites. Therefore, the forming quality of the composite material is an external factor affecting the curing deformation, which must be guaranteed in the process of part manufacturing. However, there is little research on the effect of forming quality on curing deformation.

In the meantime, resin transfer molding (RTM) can not only produce parts quickly and reduce labor costs, but it can also obtain a class A surface on both surfaces, which makes it a prime candidate for use in the high-volume manufacturing industry [20,21]. This molding method has great advantages in producing parts with low fiber volume content. When the fiber volume fraction of the product increases, the permeability of the preform will decrease significantly, which will not only increase the injection time but can even cause internal voids in the product, thus affecting the molding quality and reducing the yield of the part. On the other hand, if the size of the part is too large, or the fiber permeability is too low, it may not be possible to complete the mold filling process before resin gelation. Therefore, it is necessary to improve the RTM process to reduce resin-filling time. In recent years, several improvements have been made to the RTM process in order to shorten the molding time or increase the fiber volume fraction. One efficient approach is to incorporate compression methods into RTM, a technique known as compression resin transfer molding (CRTM). Chang [22] simulated the effect of CRTM on the resin injection of a two-dimensional model, and the results showed that the CRTM process shortened the mold filling time by 68–76% compared with the RTM process. Wen [23] experimentally compressed the fiber reinforcement in the thickness direction, and the CRTM method reduced mold filling time by 37–46% compared with RTM. Subsequently, a process combining bag compression with RTM was developed to reduce cycle times and improve the quality of the produced parts. Compared with typical RTM, the optimal vacuum-assisted process shortened the filling time by 58% and improved the bending strength by 10% [24]. Recently, Chang et al. [25] investigated the influence of process variables such as the injection pressure and the mold–opening distance compression pressure on the quality of CRTM products. They reported that compression pressure is an important variable in improving the mechanical properties of parts. From the above analyses, it can be seen that CRTM can not only significantly shorten the filling time but that it also plays an important role in improving the forming quality parts.

To date, the force applied to the mold during the CRTM process has been concentrated in a single direction on a flat surface, which is generally used to produce composite parts with simple shapes. Although plate shells have a certain effectiveness in practical applications, most applications are non-planar geometric components. Therefore, single-direction compression RTM cannot meet the practical production needs of some fields. Therefore, according to the shape and structure of the composite shell, this paper puts forward a multi-directional compression RTM (M-CRTM) method with an adjustable resin injection gap, which means that the resin injection gap can be adjusted and the fiber volume fraction can be reduced during the resin-filling process. In this paper, the RTM and M-CRTM methods were used to study the effect of porosity on curing deformations, and the deformation types of the composite shells produced by the two methods were compared. By analyzing the porosity, void size, and number of voids, the issues behind the differences in curing deformations are also discussed.

## 2. Preparation of a Composite Helmet Shell

### 2.1. Materials and Techniques

The helmet shell reinforcement was made of 300 gsm carbon fiber satin fabric, which was purchased from Tianniao High Technology Co., Ltd. (Wuxi, China). The thickness of the helmet shell was 1.2 mm, and it was arranged in a symmetrical stacking order with the layering order of [90°/0°/0°/90°]. The resin used was TDE86# epoxy resin with a viscosity of 200 mpa·s, provided by Tianjin Jingdong Chemical Composites Co., Ltd. (Tianjin, China). The curing agent was HK-021, purchased from Wenzhou Qingming Chemical Co., Ltd. (Wenzhou, China). The RTM injection machine used was a two-component precision epoxy RTM injection machine from Beijing Hengjixing Technology Co., Ltd. (Beijing, China). The resin injection process adopted the equal flow resin injection method, and the resin volume flow rate was 200 mL/min. Finally, the sample was heated at a rate of 2 °C/min and kept at 90 °C for 120 min, followed by heating at a rate of 2 °C/min and kept at 140 °C for 120 min, and finally cooled down with cooling rate of 2 °C/min.

### 2.2. Helmet Specimen Manufacturing Process

The thickness of the composite helmet prepared in this paper was only 1.2 mm. Therefore, in order to achieve the stiffness and strength required and to reduce the curing deformation for the helmet’s shell, the fiber volume fraction of the thin shell was as high as 58.8%. In this paper, the RTM resin injection process was used to produce the helmet because the surfaces of products produced by this method are smooth. The results showed that the overall deformation was large after demolding, which increases the difficulty of helmet assembly. The reason for this is that the fiber volume fraction of the product was relatively high, and due to the closed mold forming, dry spots and tiny bubbles were formed in the thin shell. To minimize these defects, a M-CRTM injection method with an adjustable injection gap was proposed in this paper. By increasing the injection gap during resin infusion, the permeability of the fabric can be improved, thus improving the overall forming quality. The schematic diagram of the forming method is shown in Figure 1.

The structure of the helmet shell is shown in Figure 1a. Firstly, the reinforced fabric needs to be evenly spread on the surface of the core mold. It can be seen that the structure of the helmet shell is a complex, special-shaped, curved surface, which adds great difficulty to its preparation. In addition, the helmet shell was designed with a mounting platform (the area marked by the red ellipse) for its later assembly with other parts. Figure 1b shows the positioning frame of the female molds, which is an important factor in adjusting the resin injection gap and in realizing the spatial positioning of the core mold. Before installing the female mold, it was first necessary to install the core mold inside the positioning frame to realize the spatial positioning of the core mold and the positioning frame. Then, the female mold was positioned in space with the core mold through the positioning frame, forming a cavity in the mold where the helmet shell will be formed. Figure 1c is the cross-sectional view of the pre-assembly of the female mold. In the pre-assembly process of the female mold, the distance between the female mold and the positioning frame was controlled to maintain a fixed distance from the pre-formed surface, allowing for control over the resin injection gap and an increase in the permeability of the fabric during the injection process. Subsequently, the prelayed fabric was injected with a two-component precision epoxy RTM injection machine. After the injection process was completed, the female mold was compressed and locked (as shown in Figure 1c) to compress the preform to the required thickness and discharge the excess resin. Finally, the female mold was tightened and heated to obtain the desired composite helmet shell. In order to study the forming quality on curing deformation, three groups of composite helmet shells were manufactured in this paper using the RTM and M-CRTM techniques and the same core mold.

### 2.3. Micro-CT Detection

The high-performance Linac CT -Diondo was used to detect the internal void distribution of the produced helmet shells. Scanning was performed using a high-resolution, small-focus CT detection system. The X-ray voltage was 150 KV, and the scanning mode was a standard cone-beam CT scan. CT detection can intuitively observe the product’s internal defects, such as defect type, location, size, etc. Current limitations of micro-CT technology mean that only small samples can be scanned. Therefore, it was necessary to cut the sample to a scannable size to estimate the void content. Three areas (areas a–c in Figure 2) were taken from each helmet shell for porosity measurement, and an internal defect map was generated after scanning.

### 2.4. Data Extraction and Curing Deformation Evaluation

As previously mentioned, the helmet shell is a special-shaped, curved structure unlike the flat plate or L-shaped models, which can be used to directly compare the warpage or spring-in. Therefore, a method is needed to quantify the curing deformation of complex-shaped structures. Reverse engineering can be used to obtain a surface point cloud of the model through 3D scanning technology, and then reverse engineering software can be used to process this point cloud to obtain the surface shape of the tested sample. Therefore, reverse engineering is often used to detect the deformation between the manufactured sample and the original model. For example, Xiang [26] used a 3D laser scanner to scan the helmet surface and compared the data with the CAD model to evaluate its deformation. In this paper, the Zero-Scan optical projection 3D scanner was used to obtain the surface profile of the composite helmet used for curing deformation analysis. The Zero-Scan optical projection 3D scanner is a non-contact full-field 3D optical measuring device with a measuring accuracy of 0.02 mm. In the scanning process, the sensor calibration and point cloud data acquisition are completed by PC-DIMIS software. Since denser point clouds can create more accurate models, the total point cloud data of each helmet housing in this paper amounted to at least 6 million points. The scanned point cloud data were then imported into the Geomagic^®^ software (3D Systems^®^) in bin format for filtering and noise reduction, followed by optimization and model reconstruction. Then, the reconstructed model was compared with the helmet design model to obtain the curing deformation of the helmet shell. The specific measurement steps are shown in Figure 3.

## 3. Results and Discussion

### 3.1. The Overall Deformation Analysis of Helmet Shell

According to the measurement method shown in Section 2.4, this paper compares the curing deformation of the three groups of helmet shells. The overall curing deformation results of the helmet shell are shown in Figure 4. Figure 4a,c,e show the helmet shell prepared by the RTM method, and Figure 4b,d,f show the helmet shell prepared using the M-CRTM technique. It can be seen that although the manufacturing technique is the same, the curing deformation of the shell is slightly different. The reason for this result is that the structure of the helmet shell presents complex characteristics, and its processing technique has a partial influence on the curing deformation.

To further characterize the curing deformation, the maximum positive and negative deformations of the shell were statistically analyzed, and the results are shown in Table 1. It can be seen from Table 1 that the average values of the maximum positive and negative deformations of the helmet prepared by the RTM process were 1.7536 mm and −1.4895 mm, respectively, which are larger than the maximum positive (1.3283 mm) and negative (−1.0085 mm) deformations of the shell formed by the M-CRTM technique. At the same time, it can be found that the maximum positive and negative deformations of the shell prepared by the two molding methods were somewhat different, and the curing deformation difference of the RTM method was larger than that of the M-CRTM method, which is caused by different preparation processes. On the other hand, by comparing the curing deformation of each group of helmet shells, it can be found that the curing deformation of the composite helmet manufactured by the M-CRTM technique was smaller than that prepared by the RTM method. The results show that the proposed M-CRTM technology can reduce the overall curing deformation of the sample.

According to the above analysis, although the preparation process has a partial impact on curing deformation, the overall comparison shows that the use of the newly proposed M-CRTM technology can reduce the generation of curing deformation, indicating that the influence of the manufacturing technique is the main factor influence curing deformation under the studied conditions. Therefore, in this study, other influencing factors can be regarded as recurrent overall. In order to explain the influence of forming quality caused by the manufactured technique on the curing deformation of the helmet shell, the first group of helmet shells was selected as the representative to study and analyze the curing deformation types.

Figure 4a,b show the first group of shells prepared by the RTM and M-CRTM methods, respectively. The area marked in blue is the position where the curing deformation of shells prepared by M-CRTM decreases compared to those prepared by RTM. The maximum positive and negative deformations of the RTM-molded shells were 1.6006 mm and −1.6001 mm, respectively. The maximum positive and negative deformations of the helmet shells fabricated by the M-CRTM method were 1.3817 mm and −1.0621 mm, respectively. The maximum positive deformation of the M-CRTM method was reduced by 13.7% and the maximum negative deformation was reduced by 33.6% compared to the RTM method, indicating that there is greater deformation in some areas of the RTM-molded shell. The reason for this difference is that the thickness of the helmet shell is thin, and a small change in the number of voids can affect the stiffness of the relevant areas of the shell. When there is a large internal void in these areas, the overall stiffness of the shell decreases [15], resulting in a weak resistance to deformation and an increase in the degree of curing deformation. Moreover, when there are inconsistencies in the stiffness on either side of the helmet, deflection deformation may occur in the shell. To further explain the influence of voids on the curing deformation of the helmet shell, the void distribution inside the shell was subsequently measured.

### 3.2. Void Analysis Using Micro-CT

Figure 5 shows the void distribution inside the helmet shell (area a–c in Figure 2) obtained using CT detection technology. Among them, Figure 5a,c is the helmet samples obtained using the RTM method, and Figure 5d,f is the internal defect distribution of the helmet samples obtained using the M-CRTM method at the corresponding position. By comparing Figure 5a,d, it can be seen that the internal defects of the RTM molded shell are widely distributed and there are many small voids, while the number of voids in the corresponding areas of the M-CRTM molded helmet is significantly reduced. It can be seen from Figure 5b that there are large voids inside the sample, and the voids are elongated and arranged parallel to the fiber direction (marked with red ellipses). This is because the fiber volume fraction is large and the flow velocity of resin in the fiber bundles is lower than that between the fiber bundles, resulting in the incomplete infiltration of the fiber fabric. In comparison, the proposed M-CRTM method (Figure 5e) significantly reduces the generation of large internal defects. This is because the decrease in the fiber volume fraction during the injection process can increase the permeability of the fabric, making it easier for large air bubbles to be discharged from the mold. Figure 5c shows the RTM molded sample, and it can be seen that the internal voids are randomly distributed inside the sample with a wide area. Figure 5f shows the M-CRTM-molded sample: it can be seen that the content of voids inside the sample is very small. By comparing Figure 5c,f with the internal void defects at the same position, it can be seen that the M-CRTM-molded sample greatly reduces the content of small internal bubbles. Through the comprehensive comparison and analysis of the internal voids of the two forming methods, it can be concluded that there are many large voids distributed along the fiber bundle in the RTM molding sample, and the proposed M-CRTM molding method not only reduces the generation of large voids, but also significantly reduces the number of small voids. The reason for this result is that due to the high fiber volume fraction of the helmet shell, the fabric is compacted during the mold locking process, which makes the yarn flatter and denser and results in a reduction in the distance between the fabric bundles. As a result, the resin basically flows in the fiber bundle. Moreover, due to the low permeability of the fabric, the resin cannot completely saturate the fiber bundle, resulting in the formation of large voids along the fiber bundle. Due to the reserved injection gap, the M-CRTM method can significantly reduce the fiber volume fraction during the resin injection process and enhance the fabric’s permeability. Therefore, the residual air bubbles in the preformed fiber bundles can be discharged more easily with the resin, thereby reducing the internal voids and improving the overall forming quality of the helmet shell.

Subsequently, a statistical analysis of the size distribution of the void volume inside the helmet shell was performed, and the results are shown in Figure 6. Figure 6a shows the size distribution of micro-voids for the RTM-molded helmet shell, and the total void content is 0.224%. As can be seen from the figure, the interior void volume of the sample is widely distributed, with a total void volume of 436.202 mm^3^ and a maximum void volume of 6.6mm^3^. The volume of most voids was below 2 mm^3^, and the number of voids above 2 mm^3^ was 22, accounting for 17.75% of the total void volume. Figure 6b shows the size distribution of the M-CRTM-molded helmet shell with a porosity of 0.142%, a total void volume of 276.394 mm^3^, and a maximum void volume of 4.574 mm^3^. The volume of most of the voids in the M-CRTM molded helmet was below 1 mm^3^, and the number of voids with a volume size greater than 2mm^3^ was 11, accounting for 11.54% of the total void volume. Compared with the RTM-molded helmet, the porosity of the M-CRTM-molded helmet was reduced by 36.6%, and the number of voids was significantly reduced. In addition, in the area (Figure 5b) where the curvature of the shell changes greatly, large voids will be generated because the air bubbles are not easily discharged from the mold. Large voids do not only affect the mechanical properties of manufactured parts [15] but also affect the stress distribution of cured composites, and even cause increased stress concentration [18]. However, the number of large voids in the composite helmet shell prepared by the M-CRTM process is significantly reduced. This is because the composite helmet shell reserved a 0.5 mm gap during the injection process, which reduced the fiber volume fraction from 58.8% to 41.5%. This improvement significantly increased the fabric permeability and improved the forming quality of the composite helmet.

### 3.3. Void Analysis of Helmet Shell Section

Due to the thinness of the helmet shell manufactured in this paper, the presence of small voids may have a great influence on the curing deformation of the shell. In order to more clearly characterize the voids’ positions, this paper characterized the voids in the thickness direction of the helmet shell (partial areas of Sections 1–3 in Figure 2), and the results are shown in Figure 7. By comparing the two molding methods, it can be found that bigger and more elongated voids appeared in the samples prepared using the RTM molding method. Figure 7a–c show the void distribution in the thickness direction of the helmet shell obtained using the RTM method. Although the overall porosity of the helmet shell was small, it can be observed that the helmet shell obtained by RTM molding had large voids in the thickness direction, and the size of these voids can even reach or exceed half of the thickness of the shell in some places (Figure 7a,b; red ellipses mark locations). These positions in the thickness direction of the helmet are stress concentration areas [18], and due to the presence of such large defects, the strength and modulus of this position may decrease with the change in void size to different degrees. Figure 7d,f exhibit the distribution of internal voids in the thickness direction of the helmet manufactured by the M-CRTM method at the same positions. By comparing Figure 7a,d, it can be seen that the void size decreased significantly in the thickness direction. Figure 7b,e show the void size in the area where the curvature of the helmet changes greatly. As can be seen from the figure, due to the use of a reserved resin injection gap, the number of voids was significantly reduced. Figure 7c,f show the void size of the area in the middle part of the helmet shell where the curvature changed slowly. It can be seen that the small voids in the interior were significantly reduced, and some were so small in some areas that they could not be detected by CT. In comparison to the RTM method, it can be seen that the number of large voids in the thickness direction significantly decreased, but there were still some randomly distributed small voids. The results prove that the improved injection method can improve the forming quality of the part, but the voids cannot be completely eliminated. The reason for this result is that the RTM method cannot completely infiltrate the fiber fabric in the thickness direction, and a non-infiltrating area is generated in the inner part due to the large fiber volume fraction. The proposed method reduces the fiber volume fraction during injection, making it easier for the resin to penetrate the reinforcement. Since the thickness of the composite shell is ultra-thin, with a thickness of only 1.2 mm, the formation of voids in the thickness direction may affect the thermal, mechanical and thermo-mechanical properties of the composite and further affect the curing deformation of the component [11]. Therefore, the measurement of the positions and sizes of voids plays an important role in studying their effect on the curing deformation properties of composite parts.

### 3.4. Flexural Modulus Analysis

In order to characterize the voids on the mechanical properties of composite materials, the platform parts of the helmet shells produced by the two methods were cut to test their flexural performance. However, due to the limitation in shell size, only four flexural samples could be obtained for each type of helmet. The results are shown in Table 2. It can be seen from Table 2 that the flexural modulus of the M-CRTM sample was larger than that of the RTM sample. The flexural modulus distribution of the M-CRTM sample is relatively average, and the coefficient of variation (CV) is small. However, the distribution span of the flexural modulus of the RTM sample was large, and its coefficient of variation was 2.89%, which is larger than that of the M-CRTM specimen (1.20%). Through the above analysis, it can be concluded that an increased number of voids will cause the modulus of the composite shell to decline to a certain degree. The flexural stiffness distribution of the composite shell will be more uniform by improving the forming quality.

### 3.5. Local Stress Distribution of Sample with Voids

In order to study the effect of local voids on the stress distribution of composite materials, a tensile test sample was obtained from the helmet prepared using the RTM method. Firstly, a CT scanning experiment was carried out to obtain the distribution of internal voids, and then the digital image correlation (DIC) system was used to capture the strain field of the sample in the tensile test. The results are shown in Figure 8. In Figure 8a, the red and green areas are the distribution locations of large voids (the red ellipses mark locations), and the blue areas are the distribution locations of small voids. Figure 8b shows the strain field distribution of the corresponding sample during the tensile process. It can be seen that stress concentration occurs at the locations of large voids, while the location of small voids has little effect on the stress distribution. The above results demonstrate that the voids in the helmet sample will cause uneven stress distribution and an even stress concentration.

### 3.6. Curing Deformation Analysis of Helmet Section

#### 3.6.1. Deformation Analysis of Helmet Mounting Platform

The helmet platform (the red ellipse marks in Figure 1a) will be equipped with optical components, so the deformation distance of the mounting platform has an important impact on assembly. To clearly characterize the deformation size of the mounting platform, the cross section of the platform was measured and characterized in this paper. Figure 9a–c show the deformation distribution of the cross-section (the green line in Figure 2) for the RTM-molded sample. It can be found that the deformation of the middle area of the helmet shell was less than 0.3 mm, while the deformation of the edges on both sides increased significantly. Figure 9a is an enlarged view of the part marked by the red circle on the left side of Figure 9b. It can be seen that the left part of the helmet mounting platform had a tendency to deflect outward from top to bottom, and the offset distance was shifted from 0.7907 mm to 1.2468 mm, reaching 0.4501 mm. This may be due to the internal defects in the part that reduce the rigidity in the nearby area, resulting in large deformation of the shell. Figure 9c shows the deformation distribution in the right part of the helmet platform. Different from the deformation on the left side, there is a downward trend in deformation from top to bottom, but the change is relatively small. This may be caused by torsion of the shell due to unbalanced internal stress distribution. Figure 9e shows the deformation distribution of the platform section for the M-CRTM molding sample, showing a trend of small deformation in the middle area and of large deformation at the edges on both sides, which is similar to the RTM molding sample. Figure 9d,f show the enlarged views of the platform section deformation on the left and right side of the M-CRTM-molded sample, respectively. It can be seen from the figures that there was a tendency of outward deflection from top to bottom. The reason for this result is that the stress relief in the edge region was relatively sufficient, resulting in a large deformation of the corresponding edge region. Compared with the mounting platform deformation of the RTM-molded sample, it was found that the deformation on both sides of the platform formed by M-CRTM was relatively uniform and there was no torsion phenomenon. In addition, the maximum platform deformation distance of the M-CRTM-molded shell also decreased from 1.2468 mm to 1.0061 mm: a decrease of 19.31%. This indicates that the porosity will affect the curing deformation type of the helmet shell and that the curing deformation can be reduced to a certain extent by improving the molding process and reducing the internal porosity of the parts.

#### 3.6.2. Deformation Analysis of Helmet Shell Section

In order to further study the tendency and mechanism of curing deformation, the cross-section curing deformation of the helmet shell was studied in this paper. Firstly, three representative sections (as shown in Figure 2) were selected to analyze their deformation laws, and then the 3D coordinates of points on the sections were extracted to analyze the deformation modes of the sections.

Figure 10 shows the deformation distance along the midline (Section 1) of the helmet shell. It can be seen that the RTM-molded helmet shell had the largest positive deformation (0.133 mm) and the largest negative deformation (−0.298 mm) along the midline section, which were larger than the maximum positive deformation (0.064 mm) and the maximum negative deformation (−0.217 mm) of the M-CRTM-molded helmet. This may be due to the fact that the internal defects of the RTM-molded helmet shell reduced the stiffness of the parts and the deformation resistance of the sample, resulting in the specimen being more prone to shape changes after demolding. In addition, both molding methods showed a trend of positive deformation in the middle and negative deformation at the edges on both sides. The reason for this trend is that the shape change at the edge positions on both sides was large due to the release of internal stress. Figure 10b shows the components of the deformation in the X, Y, and Z directions for this cross-section. It can be seen that the deformation along the central line was small in the X direction, indicating that the deformation in this section had little relationship with the X deformation. At the same time, the curing deformation on both sides mainly came from the deformation in the Y direction, which was also the main reason for the negative deformation. In addition, the positive deformation in the middle region mainly came from the Z direction, which was the main reason for the positive deformation. By comparing the curing deformation displacement of the two types of helmets, it was found that the M-CRTM method obtained a smaller curing deformation in the Y direction and the Z direction. This proves that the proposed molding process has a positive effect on improving the overall shape retention of the helmet shell.

Figure 10 is the curing deformation distance diagram at different positions of Section 2. In this figure, 400 points were equally selected along the section to analyze the curing deformation trend at this position. The position of the midline (Section 1) was taken as the 0 point, the point on the left was from −200 to −1, and the position point on the right was from 1 to 200. It was specified that the outward deformation is a positive value and the inward deformation is a negative value.

As can be seen from Figure 11a, the deformation of the RTM-molded helmet at the position of Section 2 presented asymmetry and tended to shift from the left side to the right side, indicating that a certain degree of deflection deformation occurred after the helmet is released from the mold. This may be due to the uneven distribution of internal defects (as shown in Figure 5b) in the RTM-molded helmet sample, which makes the stiffness different in each region of the sample, leading to an uneven distribution of internal stress. In comparison, the deformation of the helmet formed using M-CRTM had a small difference between the two sides, indicating that by reducing the internal porosity, especially the generation of large voids (as shown in Figure 5e), the internal stress distribution of the part can be made uniform and the curing deformation can be reduced. In order to further study the deformation form of the helmet shell, we studied the deformation components of the shell in the three directions of X, Y, and Z, as shown in Figure 11b,d. Contrasting the variation trend of the deformation components of the midline (Section 1), deformation occurred in all directions for Section 2. By comparing the deformation displacement in the X direction, it was found that the deformation of the M-CRTM-molded sample in the X direction was symmetrical with respect to the midline, and the maximum deformation was similar on both the left and right sides. However, the curing deformation of the RTM-molded sample in the X direction showed a rapid increase in the left side and a slight decrease in the right side, which indicates that irregular deformation was induced in the helmet shell. In addition, by comparing Figure 11a,b, it is found that the deformation of the RTM-molded sample in the X direction is similar to the overall amount of deformation, indicating that at the position of Section 2, the deformation is mainly affected by deformation component in the X direction. Figure 11c shows the deformation distance in the Y direction of Section 2. It can be seen that there was a small difference in deformation distance between the two forming methods, except at the right edge, where the opposite trend occurred (marked in blue), caused by the overall deflection of the helmet sample. Figure 11d shows the deformation distance of Section 2 in the Z direction. It can be seen that the curing deformation of the RTM-molded shell in the Z direction of this section was high on the left and low on the right side (the blue marked area). Moreover, the maximum negative deformation of the shell was −0.228 mm, which is significantly bigger than the −0.142 mm of the M-CRTM-molded sample, indicating that the helmet shell was deformed in the Z direction at this section. The deformation component of the M-CRTM-molded shell in the Z direction had a high degree of symmetry on the left and right sides, which indicates that the curing deformation value of the helmet shell in the Z direction was relatively uniform after the process was improved. Through comprehensive comparison of the curing offsets in all directions, it was found that the displacement of the RTM-molded shell mainly has a positive displacement change in the left side and a negative displacement change in the Z direction on the right side at the position of Section 2. The reason for the abnormal displacement changes in the X direction is that the part deflects in the Z direction in the front half, resulting in the downward deflection of the right side of the helmet. The comprehensive analysis shows that the deflection deformation in the X direction mainly occurs in the front half of the helmet shell. In addition, by comparing the deformation displacements of the two helmets, it can be seen that by improving the molding process, the M-CRTM-molded samples obtained smaller curing deformations in the X, Y, and Z directions. The above results demonstrate that it is beneficial to reduce the amount of curing deformation by reducing the internal porosity of the composite shell.

The curing deformation diagram of Section 3 is shown in Figure 12. As can be seen from Figure 12a, the deformation of the RTM-molded helmet shell presented prominent asymmetry on the left and right sides. The maximum positive deformation on the left and right sides reached 0.846 mm and 1.138 mm, respectively, with a difference of 25.7%. The reason for this result is that the large difference in local porosity (as shown in Figure 4a) after curing leads to an uneven distribution of internal stress, resulting in a certain degree of torsional deformation. However, the difference in the curing deformation on both sides of Section 3 was small for the M-CRTM-molded helmet. The maximum positive deformations on the left and right sides were 1.026 mm and 0.978 mm, respectively, and the difference was only 4.7%. This result proves that by improving the molding process and improving the forming quality, the amount of curing deformation can be reduced.

Subsequently, we studied the deformation components in the X, Y, and Z directions of Section 3, as shown in Figure 11d and Figure 12b. From the curing deformation distance in the X direction shown in Figure 11b, it can be seen that when the measurement point was located at the right edge, the deformation displacement of the M-CRTM-molded sample was larger at the same position. When the measurement point was located on the left edge, the deformation displacement of the RTM-modeled sample was larger at the same position. This indicates that the RTM-molded helmet shell was more deformed in the positive X direction compared to the M-CRTM-molded sample. Moreover, by comparing Figure 12a,b, it can be seen that the deformation of the RTM-molded shell in the X direction is similar to that of the overall deformation, indicating that the deformation of Section 3 was also mainly affected by the deformation in the X direction. From Figure 12c, it can be seen that the deformation displacements of the two molding methods in the Y direction are very small, which indicates that the position of Section 3 was less affected by the deformation in the Y direction. Figure 12d exhibits the deformation distance component of Section 3 in the Z direction. It can be seen that the deformation of the RTM molded helmet shell in the Z direction of Section 3 is larger on the left side and smaller on the right side (the black marked area), which indicates that the helmet shell produced deflection deformation in the Z direction. A comprehensive comparison of the offset distance in each direction shows that the curing deformation in the left direction of Section 3 mainly occurred in the positive X direction, followed by the displacement deformation in the positive Z direction for the helmet shell prepared by RTM. That is, the X–Z deflection deformation mainly occurred in the rear of the helmet shell.

In general, by comparing the curing offset distances of the two types of helmets in the X, Y, and Z directions, it can be found that the newly proposed M-CRTM process can reduce the overall curing deformation of the helmet shell. The results show that the shape preservation of the cured shell can be effectively improved by reducing the internal porosity of the shell.

## 4. Conclusions

The existence of void defects results in large curing deformations in composite thin shells. In this paper, a multidirectional compression RTM (M-CRTM) method with an adjustable resin injection gap was proposed. By comparing the curing deformation of helmet shells manufactured by the RTM and M-CRTM methods, it is found that although the manufacturing techniques are the same, the curing deformations were slightly different. The reason for this result is that the helmet structure has complex characteristics and the processing technique used also has a partial influence on the curing deformation. The study found that the use of the newly proposed M-CRTM technology can reduce the overall curing deformation. Therefore, this paper selects a group of helmet shells as representatives to study the influence of forming quality on the type of curing deformation. The X-ray CT scan test found that there were large voids in the composite helmet prepared by RTM, even reaching or exceeding half of the shell’s thickness in some areas. In comparison, the M-CRTM-molded helmet shell not only reduced the size and number of large voids, but it also significantly reduced the internal porosity of the shell. By studying the curing deformation in the X, Y, and Z directions of three representative sections and the overall curing deformation, it was found that the warpage and torsional deformation occurred in the RTM-modeled helmet. Furthermore, the maximum positive deformation and maximum negative deformation of the M-CRTM helmet shell were reduced by 13.7% and 33.6%, respectively. The reason for this result is that when the forming quality of the shell is improved, the internal stress and stiffness of the shell are uniformly distributed, and the curing deformation is thus significantly reduced. Although the change in the degree of curing deformation takes place at a millimeter level, such a difference plays an important role in aerospace parts that require high assembly accuracy. Our proposed M-CRTM method thus has a positive effect on further improving the application of composite thin shells in industry.

In this paper, the influence of the forming quality of helmet shells on curing deformation was studied. However, this study has two shortcomings. On the one hand, void distribution models of the helmet shells were not established in this manuscript. On the other hand, due to the influence of the material model and void model, the curing deformation of the composite helmet shell was not simulated and compared with the experiment.

## Figures and Tables

**Figure 1 polymers-14-05564-f001:**
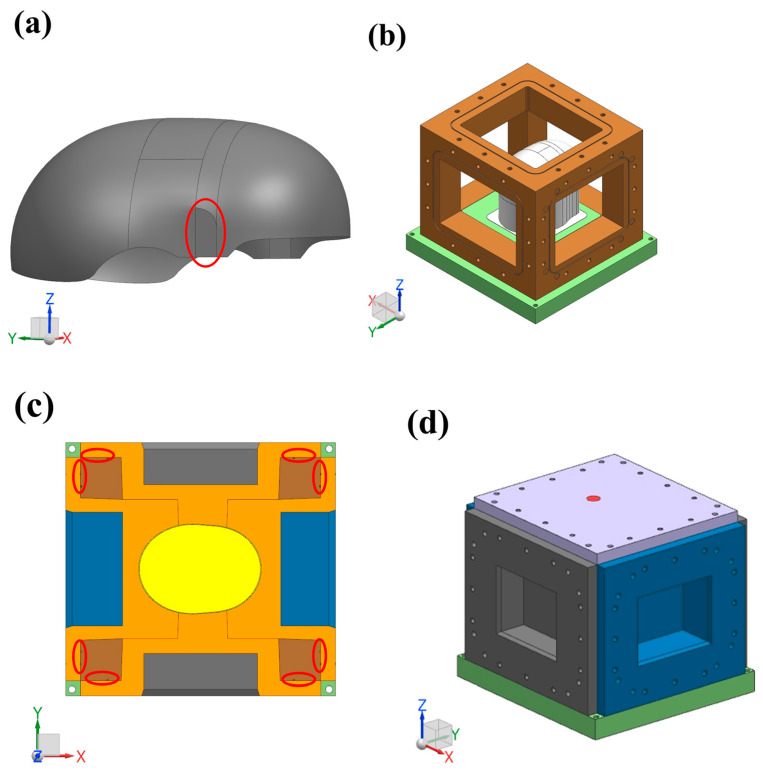
M-CRTM molding process with adjustable injection gap: (**a**) the designed composite helmet model; (**b**) the positioning frame of female molds; (**c**) cross-sectional view of female mold pre-assembly; (**d**) multi-direction compression and locking of the mold.

**Figure 2 polymers-14-05564-f002:**
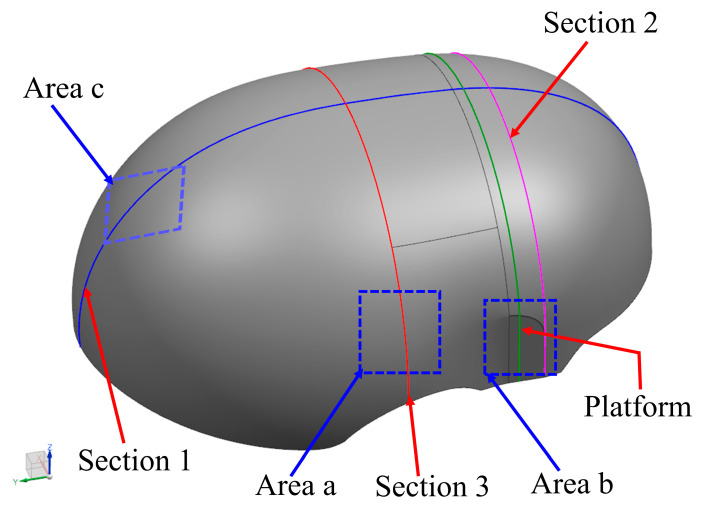
Location of CT measurement area.

**Figure 3 polymers-14-05564-f003:**
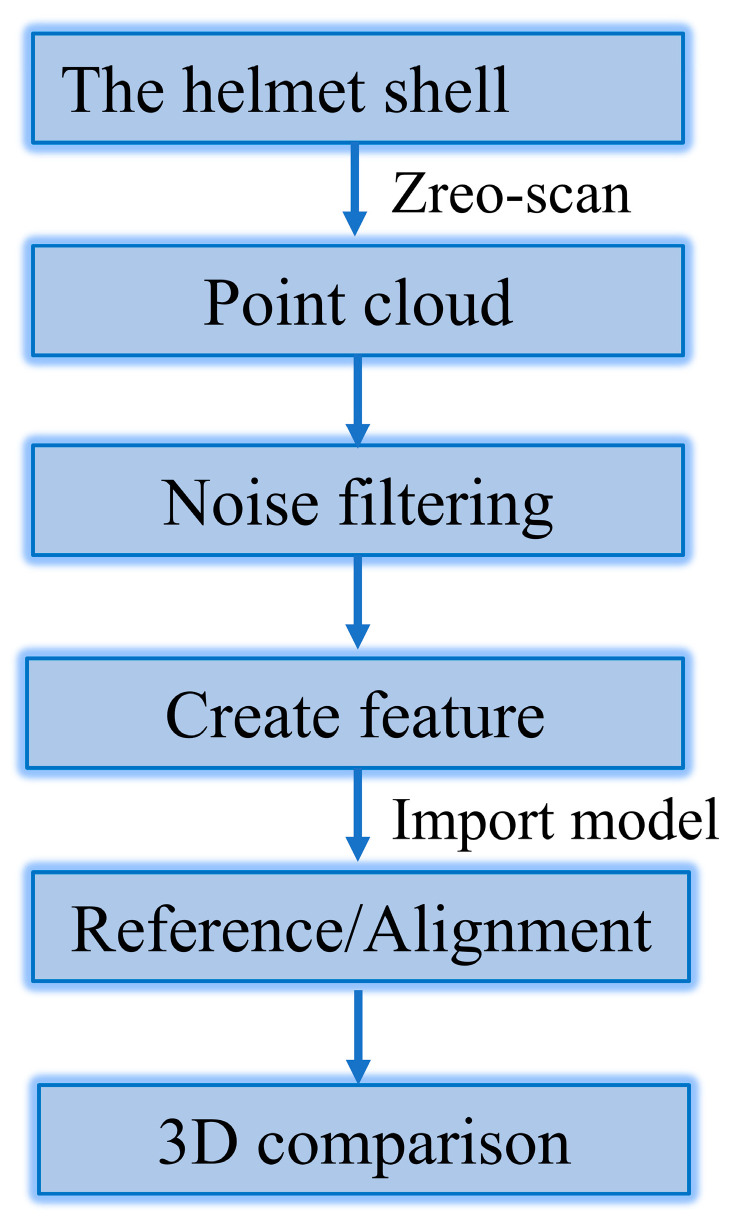
Work-flow of Zero-Scan 3D comparison.

**Figure 4 polymers-14-05564-f004:**
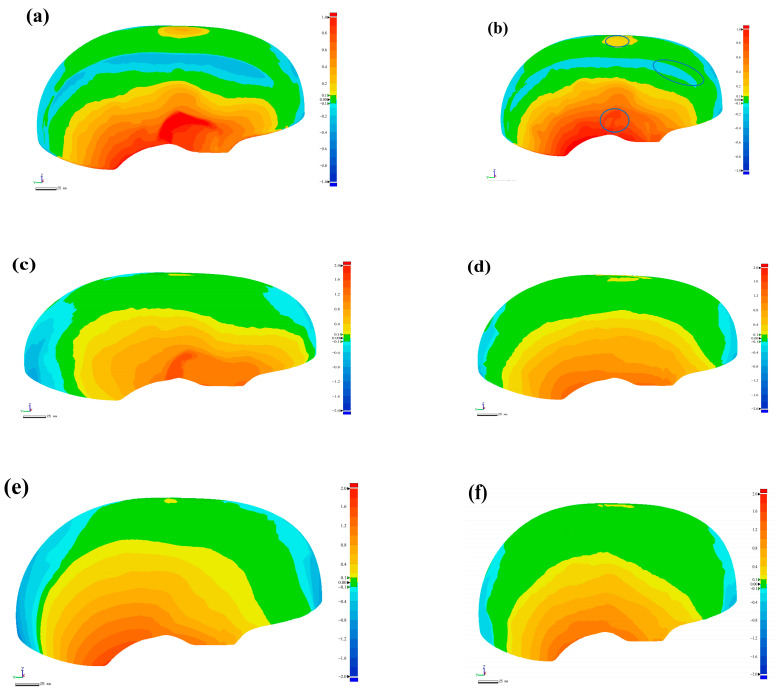
Three-dimensional profile comparison of helmets. (**a**) The helmet manufactured using RTM in the first group; (**b**) the helmet manufactured using M-CRTM in the first group; (**c**) the helmet manufactured using RTM in the second group; (**d**) the helmet manufactured using M-CRTM in the second group, (**e**) the helmet manufactured by RTM in the third group; (**f**) the helmet manufactured by M-CRTM in the third group.

**Figure 5 polymers-14-05564-f005:**
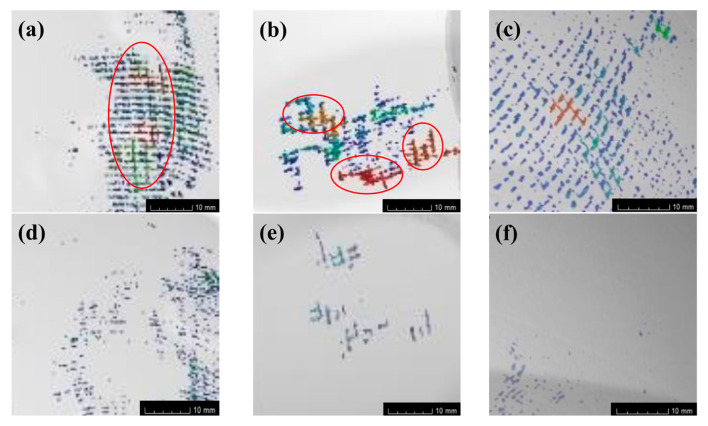
Micro-CT internal voids detection: (**a**–**c**) defects of samples prepared by RTM process; (**d**–**f**) internal defects of samples prepared by M-CRTM process.

**Figure 6 polymers-14-05564-f006:**
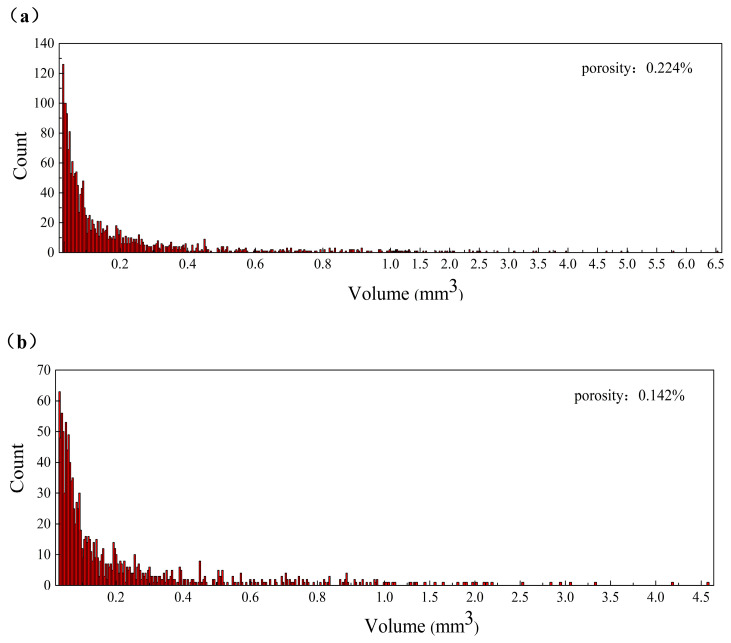
Size distribution of the voids for composite shells: (**a**) size distribution of voids for helmet prepared by RTM process; (**b**) size distribution of voids for helmet prepared by M-CRTM process.

**Figure 7 polymers-14-05564-f007:**
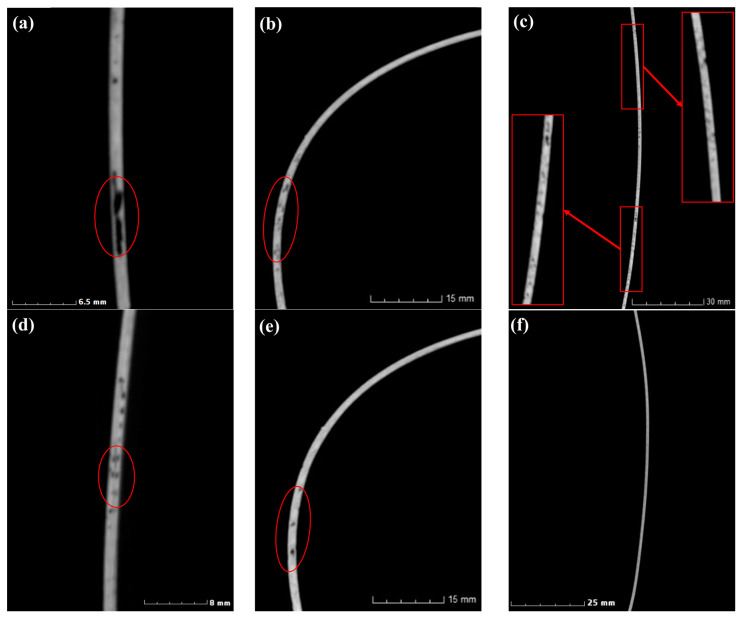
Void characterization of cross-section for helmet shell: (**a**–**c**) helmet prepared using RTM process; (**d**–**f**) helmet prepared using M-CRTM process.

**Figure 8 polymers-14-05564-f008:**
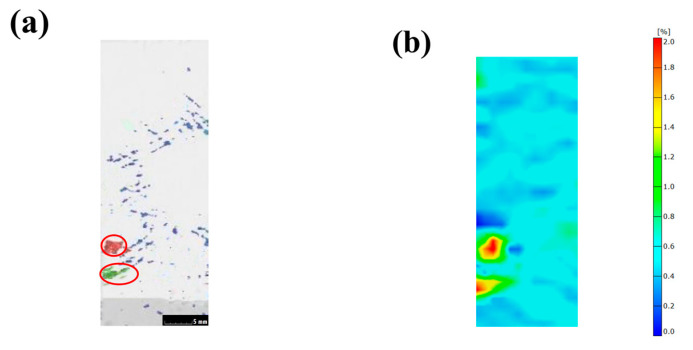
(**a**) The distribution of internal voids; (**b**) strain fields obtained by the DIC system.

**Figure 9 polymers-14-05564-f009:**
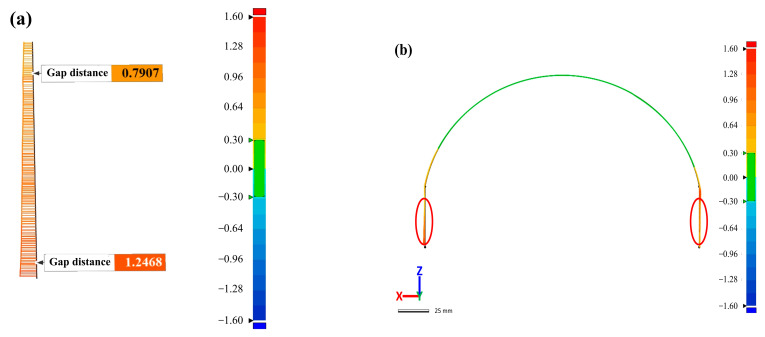
Deformation analysis of the installation platform section. (**a**) Deformation of the left side of the RTM helmet platform; (**b**) overall deformation of the RTM helmet installation platform; (**c**) deformation of the right side of the RTM helmet platform; (**d**) left side of the M-CRTM helmet platform deformation; (**e**) the overall deformation of the M-CRTM helmet mounting platform; (**f**) the deformation of the right side of the M-CRTM helmet platform.

**Figure 10 polymers-14-05564-f010:**
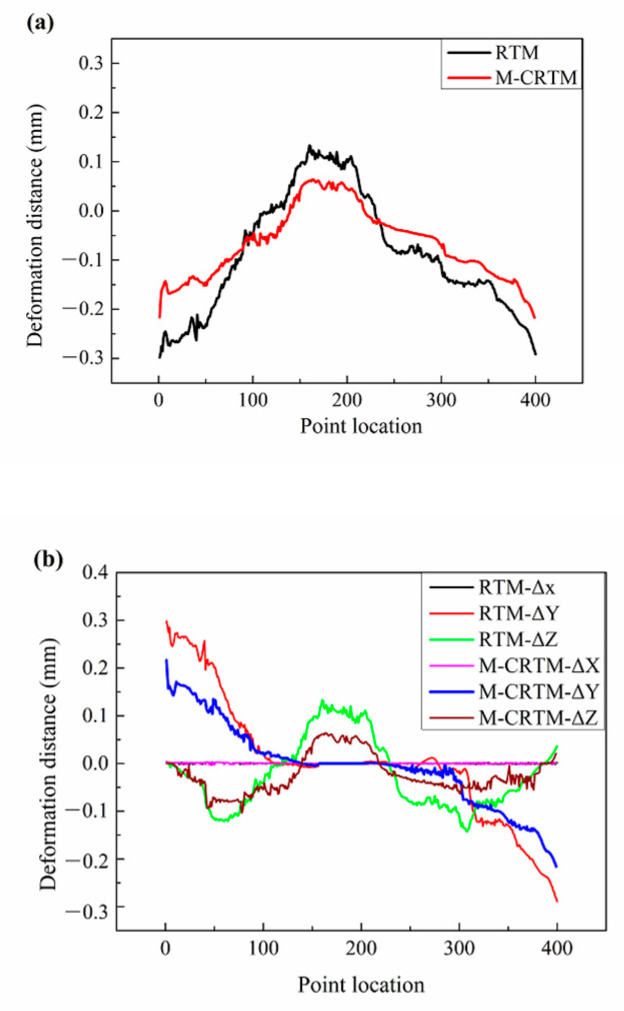
Deformation distance analysis of Section 1. (**a**) The total deformation distance of Section 1; (**b**) the deformation distance of Section 1 in X, Y, and Z directions.

**Figure 11 polymers-14-05564-f011:**
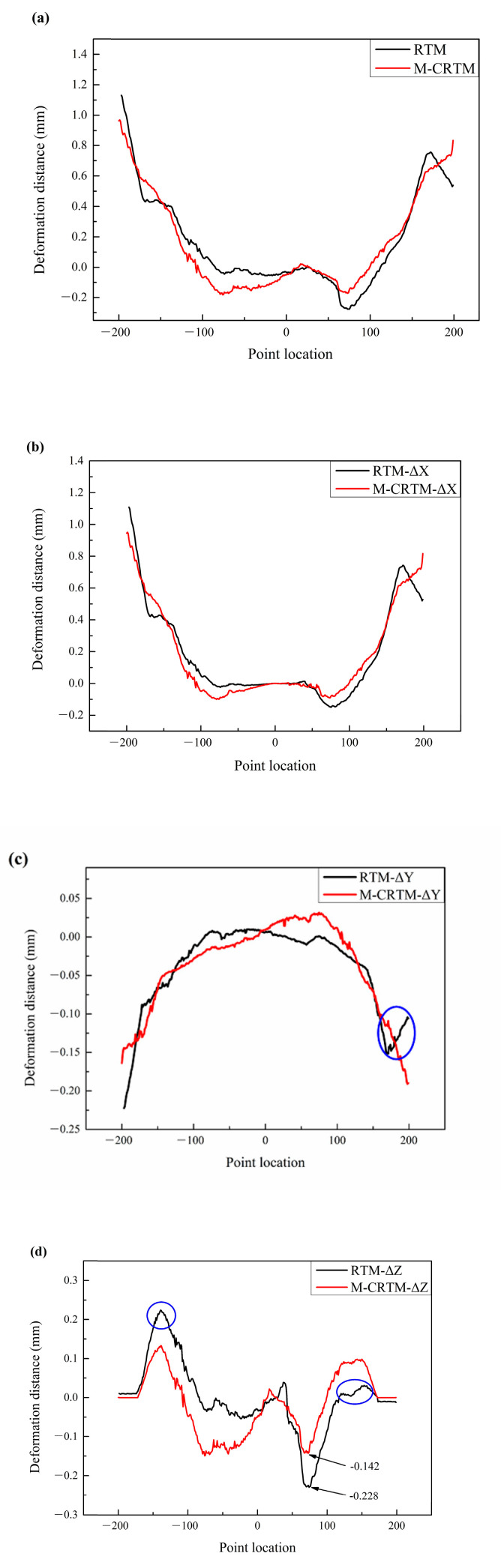
Deformation distance analysis of Section 2. (**a**) The overall deformation distance of Section 2; (**b**) deformation distance of Section 2 in the X direction; (**c**) deformation distance of Section 2 in the Y direction; (**d**) deformation distance of Section 2 in the Z direction.

**Figure 12 polymers-14-05564-f012:**
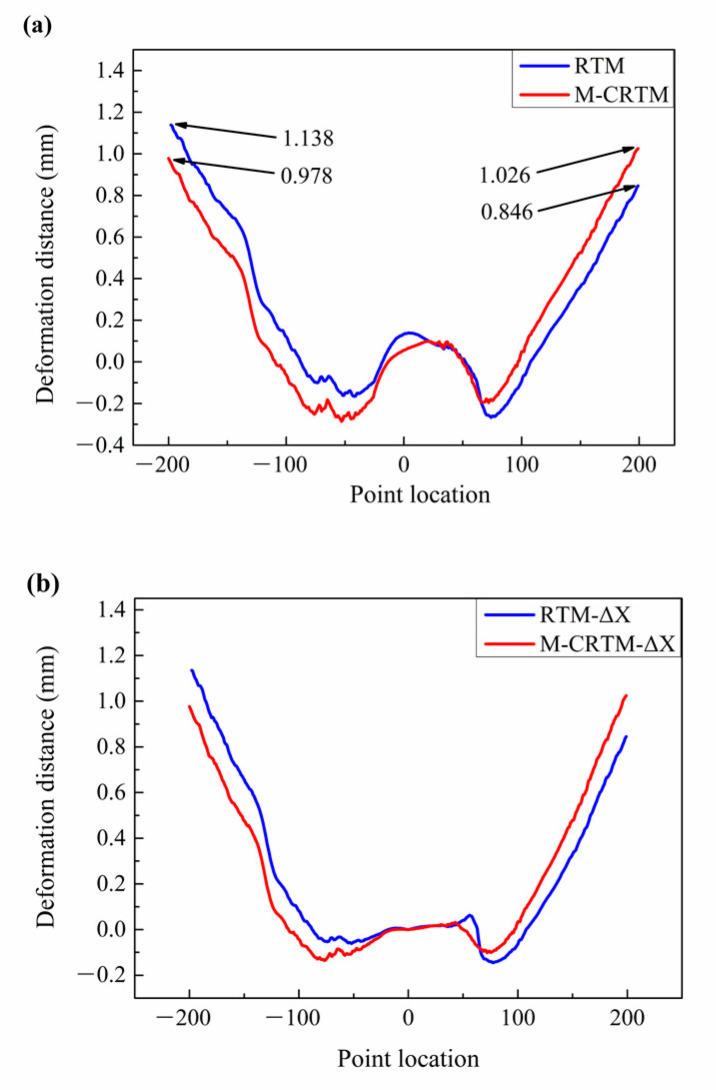
Deformation distance analysis of Section 3. (**a**) The overall deformation distance of Section 3; (**b**) deformation distance of Section 3 in X direction; (**c**) deformation distance of Section 3 in Y direction; (**d**) deformation distance of Section 3 in Z direction.

**Table 1 polymers-14-05564-t001:** Statistics of maximum curing deformation of the RTM and M-CRTM samples.

Sample ID	RTM		M-CRTM	
	MPD (mm)	MND (mm)	MPD (mm)	MND (mm)
First group	1.6006	−1.6001	1.3817	−1.0621
Second group	1.8050	−1.4163	1.2981	−0.9283
Third group	1.8551	−1.4523	1.3051	−1.0351
Average	1.7536	−1.4895	1.3283	−1.0085

Note: MPD stands for maximum positive deformation; MND stands for maximum negative deformation.

**Table 2 polymers-14-05564-t002:** Flexural modulus of RTM and M-CRTM samples.

Samples	Flexural Modulus of RTM (GPa)	Flexural Modulus of M-CRTM (GPa)
1	50.88	51.95
2	48.23	50.86
3	50.31	50.30
4	47.46	51.41
$\bar{X}$	49.86	51.13
CV (%)	2.89	1.20

## Data Availability

The data that support the findings of this study are available within the article.

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
