# Peer review of "Study on Curing Deformation of Composite Thin Shells Prepared by M-CRTM with Adjustable Injection Gap"

_polymers, 2022, doi:10.3390/polym14245564_

Round 1
Reviewer 1 Report
The paper presents a study on curing deformation of composite thin shell prepared by the M-CRTM with adjustable injection gap. This paper is interesting, however Some critical issues that the authors should carefully check before consider it for publication.
1. Figs 4, 5,8, 9, 10, 11 present low resolution. At least, must be 300dpi.
2. The deformation types of the shell should be compared with the results obtained from numerical solutions.
3. D-check English grammar.
4. The introduction must be revise, there are some publications relate to effect of porosity on composite structures such as: https://doi.org/10.1016/j.compstruct.2020.113216; https://doi.org/10.12989/anr.2022.12.5.441
Author Response
Thanks for the reviewer's advice, which is of great help to improve the quality of the manuscript. We have responded to the comments point-by-point in the revised manuscript.
Please see the attachment.

Reviewer 2 Report
The article focuses an adaptation of C-RTM to a multi-axis compression RTM scheme. Although applicable to a reduced set of components (e.g. helmets) with specific geometrical design, the technique proposed has its merits and some potential of application.
The results discussion is very difficult to follow. The authors should focus more on self-explanatory images, graphs, quantitative analysis and avoid long texts to discuss results. A good example to follow are the results shown in Figure 7. In fact, the deformation (distorsion) analysis and the impact of the processing technique on the helmet distortion is clearer and easier to follow.
The authors should also consider to quantitatively evaluate the effect of voids (and implicitly the impact of processing conditions) on the mechanical behavior of the produced components. This could be done through, for example, impact analysis of the produced helmets under different processing conditions.
The authors should also address a careful review of the text to avoid some misunderstandings such as : “When the fiber volume fraction of the product increases, the permeability of the preform will increase significantly, which…”. In fact, the permeability decreases with increase of fiber volume fraction.
Author Response

(The authors gave the same response as above.)

Reviewer 3 Report
The manuscript entiteled ‘Study on curing deformation of composite thin shell prepared by the M-CRTM with adjustable injection gap’ proposed a multidirectional compression RTM (M-CRTM) method with adjustable resin injection gap according to the shape of the thin shell. This method can increase the injection gap to reduce the fiber volume fraction during the injection process. The authors should address the following comments before the manuscript may be accepted for publication in Polymers.
1. Figure 10 and Figure 11 are not clear?
2. In conclusion ‘when the forming quality of the shell is improved, the internal stress and stiffness of the shell are uniformly distributed’. Are there any simulation or test results for the internal stress and stiffness distribution?
Author Response

(The authors gave the same response as above.)

Reviewer 4 Report
Study on curing deformation of composite thin shell prepared by the M-CRTM with adjustable injection gap
1. Introduction
Reviewer comments: Introductory part should be revise to include references that may enable further insights into the approached topic. Authors should bring forth more clear statements on their contribution and novelty. Since there several factors affecting the curing deformation, authors should point out dominant factors considered and how will be handled the correlation among them.
Recommended references:
C. Zhang, et. al. – Review of curing deformation control method for carbon fiber reinforced resin composites, Polymer Composites, http://doi.org/10.1002/pc.26648, 2022
Z. Liu, et.al. – An alternative method to reduce process-induced deformation of CFRP by introducing prestress, Chinese Journal of Aeronautics, vol. 35, no. 8, 2022, pp. 314-323, http://doi.org/10.1016/j.cja.2022.03.005
2. Preparation of composite helmet shell
2.1 Materials and techniques
Reviewer comments: Details are required on materials selection and layering, manufacturing conditions and settings, proven their influence on the parts. Authors should be able to regard as recurrent or nonrecurrent during quality assessment of their final components.
2.4 Data extraction and curing deformation evaluation
Reviewer comments: Details are imperative on the measurement device deployed for curing deformation evaluation. Proven poor description with this sub-section, additional information is needed and should enable readers further insights and traceability. Reverse engineering technique and deployment need to be addressed correspondingly and accounted in the introductory section as well.
3. Results and discussions
Reviewer comments: Discussions carried over subsections under this contribution part should bring statistically correctness. Comparisons should be carried out not only from different manufacturing techniques perspective but within individual processes, by running investigations on more than one representative helmet. Additionally, processing technique can be regarded partially responsible for the curing deformation. To be able to account only this particular issue, authors should bring forth evidence that other influencing factors (e.g., materials, composite architecture, tooling, etc.) can be regarded as overall recurrent.
Few statements need proof of evidence, references from literature or personal files.
Examples: ‘Big voids will not only affect the mechanical properties of the local area for the part, but also affect the curing shrinkage stress of the resin during the curing process’ – page 6
‘… the mechanical properties change and curing shrinkage in the thickness direction will significantly affect the deformation of the component’ – page 8
Conclusions
Reviewer comments: Author draw conclusions based on their findings. Broadening their perspective may enable them to reconsider few.
References
Reviewer comments: References can be regarded to be in relation with the topic approached. An extension article/reviews/books data base for this particular contribution may bring a widening perspective on how this hot topic was tackled by other research groups.
Author Response

(The authors gave the same response as above.)

Round 2
Reviewer 4 Report
The following statement from previous comments on the manuscript should be considered closely - ''Discussions carried over subsections under this contribution part should bring statistically correctness''.
Thank you!
Author Response

(The authors gave the same response as above.)
